

# Genomic organization, gene expression and activity profile of *Marinobacter hydrocarbonoclasticus* denitrification enzymes

Cíntia Carreira[1,2], Olga Mestre[1], Rute F. Nunes[1], Isabel Moura[2] and Sofia R. Pauleta[1]

[1] Microbial Stress Lab, UCIBIO, REQUIMTE, Departamento de Química, Faculdade de Ciências e Tecnologia, Universidade Nova de Lisboa, Caparica, Portugal
[2] Biological Chemistry Lab, LAQV, REQUIMTE, Departamento de Química, Faculdade de Ciências e Tecnologia, Universidade Nova de Lisboa, Caparica, Portugal

## ABSTRACT

**Background**. Denitrification is one of the main pathways of the N-cycle, during which nitrate is converted to dinitrogen gas, in four consecutive reactions that are each catalyzed by a different metalloenzyme. One of the intermediate metabolites is nitrous oxide, which has a global warming impact greater then carbon dioxide and which atmospheric concentration has been increasing in the last years. The four denitrification enzymes have been isolated and biochemically characterized from *Marinobacter hydrocarbonoclasticus* in our lab.

**Methods**. Bioinformatic analysis of the *M. hydrocarbonoclasticus* genome to identify the genes involved in the denitrification pathway. The relative gene expression of the gene encoding the catalytic subunits of those enzymes was analyzed during the growth under microoxic conditions. The consumption of nitrate and nitrite, and the reduction of nitric oxide and nitrous oxide by whole-cells was monitored during anoxic and microoxic growth in the presence of 10 mM sodium nitrate at pH 7.5.

**Results**. The bioinformatic analysis shows that genes encoding the enzymes and accessory factors required for each step of the denitrification pathway are clustered together. An unusual feature is the co-existence of genes encoding a *q*- and a *c*-type nitric oxide reductase, with only the latter being transcribed at similar levels as the ones encoding the catalytic subunits of the other denitrifying enzymes, when cells are grown in the presence of nitrate under microoxic conditions. Using either a batch- or a closed system, nitrate is completely consumed in the beginning of the growth, with transient formation of nitrite, and whole-cells can reduce nitric oxide and nitrous oxide from mid-exponential phase until being collected (time-point 50 h).

**Discussion**. *M. hydrocarbonoclasticus* cells can reduce nitric and nitrous oxide *in vivo*, indicating that the four denitrification steps are active. Gene expression profile together with promoter regions analysis indicates the involvement of a cascade regulatory mechanism triggered by FNR-type in response to low oxygen tension, with nitric oxide and nitrate as secondary effectors, through DNR and NarXL, respectively. This global characterization of the denitrification pathway of a strict marine bacterium, contributes to the understanding of the N-cycle and nitrous oxide release in marine environments.

Corresponding author
Sofia R. Pauleta, srp@fct.unl.pt

## INTRODUCTION

Under anoxic conditions, some organisms can use nitrate or nitrite as the sole electron acceptor in a pathway known as denitrification. This pathway allows the generation of an electrochemical gradient across the inner membrane, leading to energy conservation and ATP synthesis (*Zumft, 1997*).

The denitrification pathway is composed of four consecutive reactions, each catalyzed by a different metalloenzyme, that in different organisms can be catalyzed by more than one metalloenzyme (Table S1), and only the co-existence of *nar* and *nap*, encoding nitrate reductases, has been reported in some bacteria (*Heylen et al., 2007*). These steps are the reduction of nitrate to nitrite, by nitrate reductase (NaR), then to nitric oxide (NO), by nitrite reductase (NiR), to nitrous oxide ($N_2O$) by nitric oxide reductase (NOR) and, finally, to molecular dinitrogen, a reaction catalyzed by nitrous oxide reductase ($N_2OR$) (*Zumft, 1997*).

The genes encoding these enzymes and some of the accessory factors, involved in the biosynthesis of their active centers are located in close proximity and are sometimes transcribed as polycistronic units (*Pauleta, Dell'Acqua & Moura, 2013*; *Zumft, 1997*). The regulatory network of those genes is complex and vary between organisms, but signals as oxygen, nitrate/nitrite, NO and cell redox changes are usually implied in triggering a cascade of transcription factors able to activate or repress the promoters of denitrification genes. Although the regulatory mechanisms are not completely understood, some proteins, such as fumarate-nitrate reduction regulator (FNR), dissimilatory nitrate respiration regulator (DNR) and nitrite and nitric oxide regulator (NNR) or their orthologs, that sense those signals through iron-sulfur centers or hemes, bind DNA sequences at the promoters regions of the denitrification genes (*Spiro, 2012*).

Moreover, most of the studies on the denitrification pathway have been focused on Gram-negative bacteria that occupy terrestrial niches, using *Paracoccus denitrificans* (alpha-proteobacteria), *Pseudomonas stutzeri* and *Pseudomonas aeruginosa* (gamma-proteobacteria), as model organisms (2017; *Eady, Antonyuk & Hasnain, 2016*; *Stein & Klotz, 2011*). There are still few studies on the identification and diversity of genes involved in denitrification in other families of bacteria, especially marine bacteria, which could in part be related to a restricted number of denitrifying isolates and of complete genomes. Nevertheless, this group of microorganisms is having an increased attention in the latest years with some studies being reported recently (*Laass et al., 2014*; *Mauffrey et al., 2017*; *Mauffrey, Martineau & Villemur, 2015*).

The release of nitrous oxide to the atmosphere is in large extent due to anthropogenic activities, such as industrial and agriculture, but the natural sources (soil and ocean) also play a role, which has led to an increasing number of studies on the nitrogen cycle, and especially on the denitrification pathway. However, many of these studies were

performed on bacteria that occupy terrestrial niches, leaving marine bacteria without the proper attention. Therefore, and since a large amount of nitrous oxide released from natural sources comes from the ocean (*Bange, 2006*), there is a need to characterize the denitrification pathway of marine bacteria.

*Marinobacter* spp. are ubiquitous in marine environments, being found in different depths in the oceans and marine ecosystems (*Kaye & Baross, 2000*; *Yoon et al., 2003*) and so far, three strains of *M. hydrocarbonoclasticus* had its genome sequenced (*Grimaud et al., 2012*; *Ling et al., 2017*; *Singer et al., 2011*) (note that *Marinobacter aquaeolei* is a *M. hydrocarbonoclasticus* strain *Marquez & Ventosa, 2005*). This microorganism is proposed to have a high impact on the different biogeochemical cycles (*Singer et al., 2011*), including the N-cycle, and have a versatile metabolism in terms of carbon source and can use either nitrate/nitrite and/or oxygen as electron acceptors. *M. hydrocarbonoclasticus* is a moderate halophilic, and thus can be potentially used in wastewater plans with high saline content, for which its role as a denitrifier is crucial to remove nitrate and nitrite content from industrial sources (*Li et al., 2013*). *M. hydrocarbonoclasticus* 617 was chosen for the work presented here, given that all of the metalloenzymes from this pathway have already been characterized in our laboratory (*Correia et al., 2008*; *Lopes et al., 2001*; *Prudêncio et al., 2000*; *Timoteo et al., 2011*). The nitrate reductase is encoded by the *nar* operon, being a membrane enzyme with a molybdenum center, that receives electrons from the quinol pool (*Correia et al., 2008*). Reduction of nitrite in this organism is catalyzed by nitrite reductase cytochrome $cd_1$ (*Lopes et al., 2001*), releasing nitric oxide that is reduced by the membrane nitric oxide reductase $c$-NOR, which has two subunits, one containing a $c$-type heme, named NorC and NorB where the catalytic center is located (*Timoteo et al., 2011*). The last step of the denitrification is catalyzed by nitrous oxide reductase, which is a copper enzyme containing a CuA and a CuZ center (*Dell'acqua et al., 2008*; *Dell'Acqua et al., 2012*). These last three enzymes receive electrons from the periplasmic cytochrome $c_{552}$, forming transient electron transfer complexes (*Dell'acqua et al., 2011*; *Ramos et al., 2017*) (Fig. 1).

Here, the genome of *M. hydrocarbonoclasticus* SP17 was analyzed in detail to identify the location of genes encoding the catalytic domains of the denitrifying enzymes and of the accessory factors proposed to be involved in their biosynthesis or in maintaining the enzymes in an active state. Expression profiling of genes encoding these enzymes was performed during *M. hydrocarbonoclasticus* 617 growth in microoxic denitrifying conditions. The activity profile of the denitrifying enzymes and metabolites (nitrate/nitrite) was analyzed for cells cultured in a batch system under microoxic conditions and in sealed serum flasks. In addition, the affinity of the whole-cells for $N_2O$ was examined.

## MATERIALS AND METHODS

### Genome analysis

The genome sequence of *M. hydrocarbonoclasticus* SP17 (*Grimaud et al., 2012*), with the accession number NC_017067 (NCBI) was analyzed using bioinformatic tools to search for homologous genes encoding the catalytic subunits of the four enzymes that catalyze
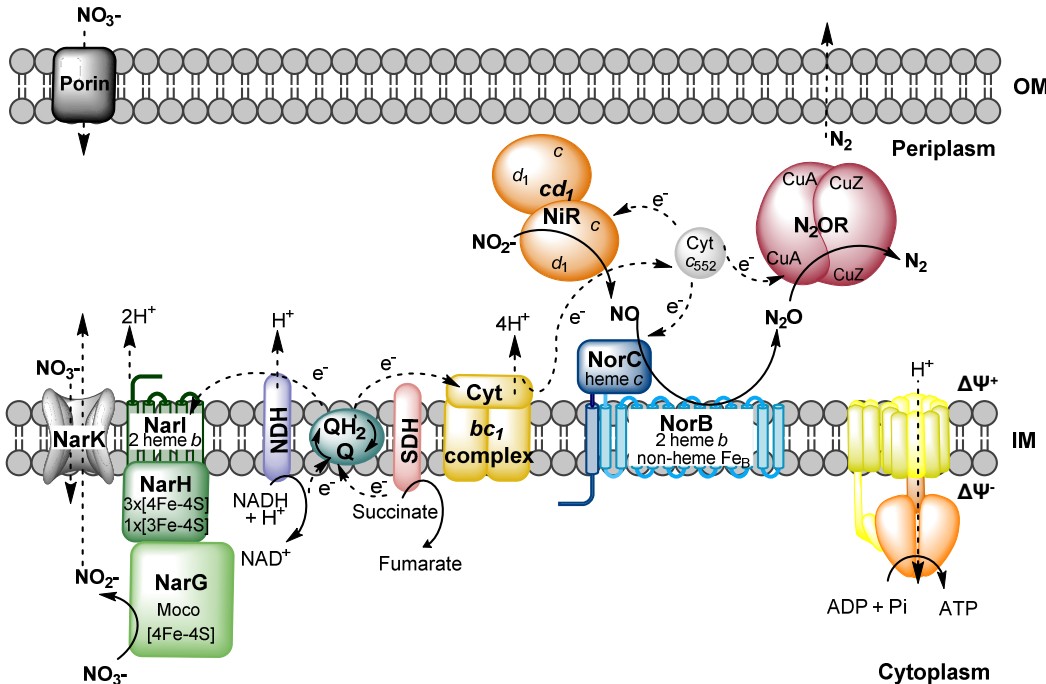

**Figure 1** **Schematic representation of denitrification pathway in *M. hydrocarbonoclasticus*.** Reactions catalyzed by those enzymes create a proton gradient across the membrane, generating a positive potential outside of the inner membrane. The electrons for each reaction come from the quinone pool. NDH and SDH are the NADH dehydrogenase and succinate dehydrogenase, respectively. Cyt is the abbreviation used for cytochrome and Moco for the molybdenum cofactor. IM and OM are the inner and outer membranes, respectively. Nitrate and nitrite type of transportation across the membrane is envisaged to occur through NarK homologues.

each step of the denitrification pathway, as well as accessory factors (see Supplementary Information S2). The virtual Footprint of PRODORIC database was used to detect the putative binding sites for FNR and IHF in the promoter regions of the denitrification genes (Table S3). As a remark, the genomic organization of denitrification genes was analyzed using the deposited genome of *M. hydrocarbonoclasticus* SP17, while the strain that was cultured was *M. hydrocarbonoclasticus* 617. In fact, *nos* gene cluster had been sequenced in our lab (GenBank: dQ504302.1) and showed to have an identical organization including *dnr* gene upstream *nos* gene cluster.

### *Marinobacter hydrocarbonoclasticus* 617 growth conditions

*M. hydrocarbonoclasticus* 617 was grown in artificial seawater (ASW) liquid medium containing 1.17% (w/v) NaCl, 0.075 % (w/v) KCl, 0.3% (w/v) $NH_4Cl$, 1.23% (w/v) $MgSO_4.7H_2O$, 0.6% (w/v) Tris, 0.1% (w/v) yeast extract and 0.12% (w/w) sodium lactate as energy and carbon source (*Baumann & Baumann, 1981*). The pH was adjusted to 7.5 with 37% (w/v) HCl. The medium was supplemented with 0.33 mM $K_2HPO_4$, 10 mM $CaCl_2.2H_2O$, 7.2 μM $FeSO_4.7H_2O$ and a Starkey oligoelement solution (*Starkey, 1938*) (1 mL/L of culture), with the composition listed in Supplementary Information S4.

*Growth in 2-L bioreactors*

In the pre-inocula required prior to the inoculation of the 2-L bioreactor, bacteria were grown under oxic conditions for 24 h at 30 °C, 210 rpm. The first pre-inoculum was prepared by inoculating 5 mL ASW liquid medium with bacteria maintained in ASW agar plates. This culture was then used to inoculate 50 mL ASW liquid medium and finally 500 mL ASW liquid medium. 200 mL of this final culture was used to inoculate 1.8 L of medium in a 2-L bioreactor. *M. hydrocarbonoclasticus* was grown in the bioreactor in a batch mode for approximately 48 h at 30 °C, using 0.75% (w/w) sodium lactate as carbon source and 10 mM sodium nitrate, as electron acceptor. The pH was continuously monitored and maintained at 7.5. A low aeration rate (0.2 vvm) and an agitation speed of 150 rpm were used during 5 h, and then it was decreased to 50 rpm. The oxygen concentration in the culture medium was monitored with a InPro 6950i G Trace Oxygen Sensor (Mettler Toledo, Columbus, OH, USA). The growth started with an oxygen saturated medium (100%) and after approximately 2 h the oxygen levels reached 0% (see Fig. S1). Cultures were visualized under an optical microscope to detect any contamination with microorganisms with a different morphology. Bioreactor growth was performed at least in triplicate. Bacterial growth in the bioreactor was monitored by collecting 4 mL aliquots into sterile vials (see Supplementary Information S5), from which optical density at 600 nm ($OD_{600 \text{ nm}}$) was measured. These aliquots were also used to quantify expression of the genes encoding the catalytic subunits of the denitrification enzymes, as well as to determine nitrite and nitrate levels and for the activity assays (see below). The aliquots were stored in liquid nitrogen until further use, after dividing the samples for each application into different sterile vials.

*Growth in sealed serum flasks*

Denitrification pathway was also analyzed in the absence of aeration at pH 7.5 using 100 mL sealed serum flasks (growth media was flushed with Argon prior to autoclave and sealed with a rubber septum and aluminium caps) containing 50 mL AWS liquid medium, supplemented as described above. The oxically grown inoculum was transferred (10% of total volume) with a syringe to the gas-tight vials. Similar conditions to the bioreactor growth were used with the following modifications: there was no aeration during the growth, the pH was continuously monitored but not adjusted (pH changed from 7.30 to 7.65) and cultures were stirred in a shaker (at 150 rpm for 5 h and then changed to 50 rpm for the remaining duration of the growth), at 30 °C. At each time-point, aliquots of 0.75 mL were sampled to measure the cell densities at 600 nm, quantify nitrate and nitrite and determine the rates of nitric oxide and nitrous oxide reduction by the cells (see below). Sampling was performed under sterile conditions.

## Nucleic acid extraction and cDNA generation

Samples of bioreactor cultures taken at different time-points were immediately frozen in liquid nitrogen, until further use (1 mL). Total RNA, from the samples, was isolated using the Isolate II RNA mini kit (Bioline, Memphis, TN, USA), according to manufacturer's instructions. Genomic DNA contamination was removed by on-column digestion with DNase I, supplied with the kit. RNA yields were determined at 260 nm and purity was

estimated by determining the $A_{260\ nm}/A_{280\ nm}$ and $A_{260\ nm}/A_{230\ nm}$ ratios. To generate cDNA, 500 ng of each RNA sample (after its extraction) was reversely transcribed using the SensiFAST cDNA Synthesis Kit (Bioline), using the following conditions: 10 min at 25 °C, followed by 15 min at 42 °C and 5 min at 85 °C (for enzyme inactivation). cDNAs were diluted (1:100) and stored at −80 °C until further use. DNA used in standard curves was extracted from a sample of the bioreactor culture taken during the exponential phase, using the Isolate II Genomic DNA Kit (Bioline, Memphis, TN, USA), according to the manufacturer's instructions and stored and −80 °C until further use.

## Quantitative Real-Time PCR

The expression of *M. hydrocarbonoclasticus* genes involved in the denitrification pathway— *narG*, *nirS*, *c-norB* (*MARHYR3054*), *q-norB* (*MARHY3014*) and *nosZ*—was analyzed by quantitative real-time PCR (qPCR). Reactions were performed using SensiFAST$^{\text{TM}}$ SYBR No-ROX Kit (Bioline) with primers specific for each gene at 250 nM and 3 μL of cDNA. Gene expression levels were determined using the relative standard curve method, which uses standard curves to interpolate unknown quantity values from a sample. This was performed using DNA standard curves, as previously described (*Gomes et al., 2005*). Briefly, for each target, as well as for the reference (16S rRNA), a standard curve was generated using serial dilutions of genomic DNA. The amount of cDNA of each target and control gene, present in every sample, was determined from their respective DNA standard curve, through conversion of the obtained threshold cycle ($C_t$) value (*Applied Biosystems, 2008*; *Gomes et al., 2005*). Relative normalized expression values were obtained by dividing the values obtained for each target gene by the values obtained for the reference gene (*16S rRNA*), as its expression is stable, during the different growth phases. Primers were designed using Primer3 software (*Untergasser et al., 2012*) (Table 1). "No template" and "RT minus" (a Reverse Transcription reaction containing all reagents except the reverse transcriptase enzyme) controls were also included in every PCR assay. Reactions were run in a Corbett Rotor-Gene instrument (Qiagen, Hilden, Germany) using the following thermal cycling conditions: 95 °C for 5 min followed by 40 cycles of 95 °C for 15 s and 60 °C for 30 s. Analysis of melting curves generated by the stepwise increase of the temperature from 60 °C to 95 °C, were used to verify the specificity of the amplified products. Three biological replicates were analyzed but not at the exact same time points, thus a single representative experiment is shown (similar results were obtained for the other replicates). Two to three technical replicates were performed depending on the biological replicates.

## Co-transcriptional analysis of *nosRZDFYL*

To determine if *nosRZDFYL* is composed of a single transcriptional unit, a sample collected during the exponential phase of *M. hydrocarbonoclasticus* growing in a bioreactor performed at pH 7.5 was used. The cDNA was generated similarly to those described above but using 2 μg of total RNA. PCR reactions were performed using 6 μL of the resulting cDNA and NZYTaq DNA polymerase (Nzytech) with primer pairs that would amplify regions between *nosR* and *nosZ* (nosR_F/nosZ_R), *nos* Z and *nosD* (nosZ_F/nosD_R) and *nosD* to *nosL* (nosD_F/nosL_R) (Table 1). Genomic DNA and a "RT minus" reactions were included

**Table 1** **Primers used in qPCR of genes encoding the denitrification enzymes and co-transcriptional analysis of the nitrous oxide reductase gene cluster.**

| Gene | Primer | Sequence (5′-3′) | Amplified fragment (bp) |
|------|--------|------------------|-------------------------|
| *narG* | narG_F | CCACCTCCTTCTTCTATGCTC | 59 |
| | narG_R | ATCCACGCCCAGTTTCTC | |
| *nirS* | nirS_F | GCGATGAAGTGTGGTTCTCT | 56 |
| | nirS_R | AACGATGGCGGATTTCTTGT | |
| *q-norB* (*MARHY3014*) | norB_F | GCTGAATACGACACCCACTC | 59 |
| | norB_R | CCAGGAAGCCATAAACACCA TGGTGTTTATGGCTTCCTGG | |
| *c-norB* (*MARHY3054*) | norB2_F | AGAAGCCCAGACCGAACT | 60 |
| | norB2_R | GCGAACACCCAGAACAGAAT ATTCTGTTCTGGGTGTTCGC | |
| *nosZ* | nosZ_R | CTGATGGCAAATGGCTGGT | 56 |
| | nosZ_F | CGGCAGGAAACGGTCTTT | |
| *nosR* | nosR_F | ACAGAAGCCGAAGATGATGC | 50 |
| | nosR_R | TGTAAACCAGGAAGCCGTG | |
| *nosD* | nosD_F[a] | GACTACCTGGGCTGGGAT | 58 |
| | nosD_R | TCGGTTCATAAGGCACATCG | |
| *nosL* | nosL_F | CTTTACCGAGCGAGAACAGA | 50 |
| | nosL_R[a] | TGTCTTTGGTGGAGCAGAAT ATTCTGCTCCACCAAAGACA | |
| *16S rRNA* | 16S_F | TAACCTGGGAACGGCATTT | 55 |
| | 16S_R | CCACTACCCTCTACCACACT | |

**Notes.**
[a] PrimerS used to amplify the intergenic region between *nosD* and *nosL*.

as controls. The amplified PCR products were analyzed by 1% agarose gel electrophoresis, run at 100 V, during 20 min, and visualized after incubation in a SYBR Safe solution (Invitrogen) under UV light.

## Nitrate/nitrite quantifications and nitric oxide and nitrous oxide reduction assays

Nitrate and nitrite quantifications along the growth curve were performed using the Nitrate/Nitrite Assay kit (Sigma), according to the manufacturer's instructions, in triplicate for each time-point. The absorbance was measured at 540 nm in a VersaMax ELISA Microplate Reader using the SoftMax Pro 6.4 software.

Whole-cell activity assays for nitric oxide and nitrous oxide reduction were performed spectrophotometrically using a TIDAS diode array spectrophotometer, at room temperature inside an anaerobic chamber (MBraun), by following the oxidation of reduced methyl viologen (non-physiologic electron donor) at 600 nm, as previously described (*Kristjansson & Hollocher, 1980*) (data analysis is described in Supplementary Information S6). Nitric oxide (5% NO/95% He, Air liquid) and nitrous oxide (>95% $N_2O$, Air Liquid) solutions were freshly prepared before the assays. Briefly, 5 mL of Milli-Q water were degassed for 30 min with argon in sealed serum flasks (sealed with a rubber
septum and aluminium caps) and then flushed with $N_2O$ for 1 h. Taking into account the solubility of $N_2O$ in water, at 25 °C and 1 atm, this solution is 25 mM in $N_2O$ (*Kristjansson & Hollocher, 1980*). The NO solution was prepared similarly but a 10% (per mass) KOH solution was used between the NO bottle and the Milli-Q solution to remove other nitrogen oxides from NO gas. A concentration of 1.91 mM in NO is reported at 20° C for a 98.5% NO stock, thus this solution is 9.7 μM (*Timoteo et al., 2011*). The assays were performed with constant stirring by adding 40 μL cell suspension to a quartz cuvette already containing 120 μM methyl viologen and 60 μM sodium dithionite in 100 mM Tris–HCl, pH 7.6 (in a final volume of 1 mL). The nitric oxide reduction assay was initiated by the addition of NO-saturated water to a final concentration of 9.6 μM, while the nitrous oxide reduction assay was initiated (without any further incubation) by the addition of $N_2O$-saturated water to obtain a 1.25 mM concentration in the assay. The reduction rate (micromoles of NO or $N_2O$ reduced per minute per optical density) was obtained by analyzing the slope of the reaction after nitric oxide or nitrous oxide addition. Controls of the reactions were performed in which nitric oxide or nitrous oxide were added to the assay in the absence of cell suspension. In these controls no oxidation of MV was observed.

The dependence of $N_2O$ in the reduction rate by whole-cells was performed in a 1 mL quartz cuvette, containing 100 μM methyl viologen and 50 μM sodium dithionite in 100 mM Tris–HCl pH 7.6, to which 3 μL of cells was added (containing 0.09 mg of total protein), collected by centrifugation at the end of the growth performed at pH 7.5 in the bioreactor, under microoxic conditions. The assay was initiated by the addition of 20, 40, 80, 120, 250 and 752 μM of $N_2O$-saturated water. Triplicates for each substrate concentration were performed. Activities were calculated as described above but reported as $\mu mol_{N2O}$ $min^{-1}$ $mg^{-1}$ of total protein (determined using the Pierce™ 660 nm protein assay with bovine serum albumin as standard). The parameters $K_m$ and $V_{max}$ were calculated by fitting the curve with the Michaelis–Menten equation.

# RESULTS

## Genomic organization of denitrification genes in *M. hydrocarbonoclasticus*

The genome of *M. hydrocarbonoclasticus* SP17 has several genes encoding proteins involved in the different steps of the denitrification pathway clustered together (see Fig. S3). The *nosZ*, encoding the catalytic domain of N2OR, is included in the *nosRZDFYL* gene cluster. Upstream the *nos* cluster is the NaR gene cluster, *narLXKGHJV* (see Fig. S3), with *narG* encoding the catalytic subunit of NaR, and *narXL*, the two-component system, NarXL, involved in transcriptional regulation of these genes in response to nitrate and/or nitrite (*Philippot, 2002*; *Spiro, 2012*).

Several nitrate/nitrite transporters have been identified in the genome of *M. hydrocarbonoclasticus*, one being associated with the *nar* cluster, other two upstream the *nir* operon, and a *narU*-type gene is also annotated in this genome. Multiple NarK-like transporters has also been identified in *P. aeruginosa* and *P. denitrificans* (*Jia et al., 2009*; *Moir & Wood, 2001*), but their biological function is still a matter of debate, with different transporters

being proposed to be required under specific growth conditions. The identification of the mechanism of action of these membrane proteins can only be accomplished through the study of the physiology of selected knock-out strains and site-directed mutagenesis.

Further upstream in the genome is *norBC*, encoding the two subunits of the short-chain membrane-bound *c*-NOR (see Fig. S3). Neither *norE* nor *norF* homologues were annotated in the genome of *M. hydrocarbonoclasticus*. However, the translated sequence of *MARHY3057* shares 46% sequence identity with *P. stutzeri* NorE (designated in that microorganism as NirQ). A BLAST search was performed using the translated sequence of *MARHY3058*, showing 53% identity with *P. stutzeri* NirP (the NorF homologue in *Pseudomonas* species with 82 residues), though in a very restricted region, covering only 49% of the primary sequence. Indeed, this putative NorF (with 61 a.a.) is 40% shorter than its homologues. However, the analysis of the entire intergenic region between *MARHY3057* and *MARHY3058* identifies a longer ORF encoding a protein with 79 residues, that has 39% primary sequence identity with *P. stutzeri* NirP. Therefore, we propose that NorF is misannotated in *M. hydrocarbonoclasticus* SP17 genome, being encoded in the 3141724–3141963 genomic region rather than in the 3141778–3141963 region.

A second *norB* gene (*MARHY3014*) also encoding a NOR is annotated in *M. hydrocarbonoclasticus* genome, downstream the *nos* cluster and transcribed in the opposite direction (see Fig. S3). This NOR is composed of a single long-chain subunit being a *q*-NOR homologue (primary sequence identity with *Geobacillus stearothermophilus* *q*-NOR is 37%, with a 95% coverage, and 25% with *P. aeruginosa* large subunit NorB, but with only 60% coverage) (Table S2). While *c*-NOR uses a *c*-type cytochrome as electron donor, *q*-NOR accepts electrons from the quinone pool. The presence of two types of NOR in the same bacteria does not seem to be frequent (*Heylen et al., 2007*; *Jones et al., 2008*) and its biological relevance is unknown.

After the *norBC* genes, a large cluster of 10 genes involved in the nitrite reduction step of the denitrification pathway was found linked together, *nirFCSDLGHJEN*. Different genetic arrangements of *nir* genes have been identified in bacteria. *P. aeruginosa* presents the most similar genetic organization of the nitrite reductase gene cluster in comparison to the one of *M. hydrocarbonoclasticus* (*Bali et al., 2014*; *Philippot, 2002*). The only differences are in the *nirM* gene, encoding cytochrome $c_{551}$, which is absent in *M. hydrocarbonoclasticus* and in *nirFC*, which are located upstream *nirS* and are transcribed in the opposite direction. NirM is the electron donor of NiR in *P. aeruginosa*, while the cytochrome $c_{552}$, encoded by *MARHY3556*, was identified as the physiological electron donor of NiR in *M. hydrocarbonoclasticus* (*Lopes et al., 2001*) and is distantly located in the genome.

The denitrification pathway has been proposed to be regulated mainly by transcription regulators of the CRP/FNR family, FNR and DNR, that in *M. hydrocarbonoclasticus* are encoded by *MARHY0862* (FNR homologue) and *MARHY3023* (DNR homologue), with this later ORF being located upstream the *nos* gene cluster. FNR, that activates genes during anaerobiosis, is a homodimer containing a [4Fe-4S] center per protomer, which is sensitive to oxygen and to nitrosylation (*Crack et al., 2014*). On the other hand, DNR senses N-oxides, in particular NO (*Arai, Kodama & Igarashi, 1999*), through a heme (*Castiglione et al., 2009*). These two transcription regulators bind specifically to consensus sequences

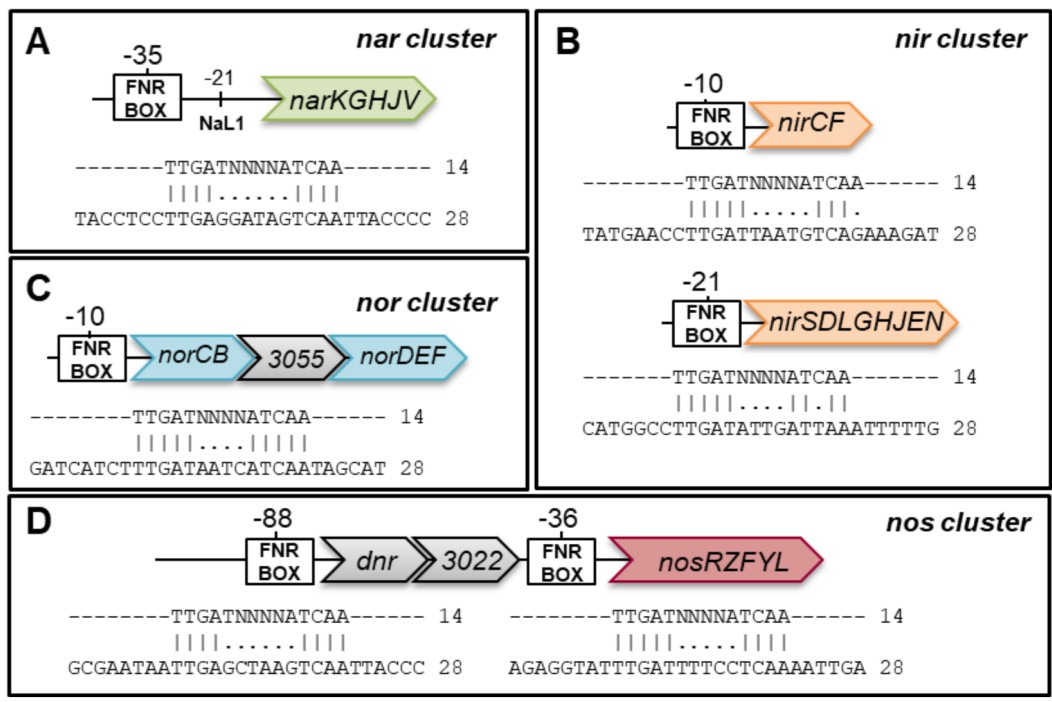

**Figure 2 Analysis of the promoter regions of nitrate reductase (nar), nitrite reductase (nir), nitric oxide reductase (nor) and nitrous oxide reductase (nos) gene clusters.** Analysis of the promoter region of (A) *nar*, (B) *nir*, (C) *nor* and (D) *nos* genes. Sequence alignments between the identified putative FNR/DNR-binding sequences and the consensus sequence are shown, as well as its position from the putative transcription initiation site. The putative NarL binding site is evidenced in the *nar* cluster. A putative FNR/DNR box was identified in the promoter region of the *dnr* gene upstream *nos* gene cluster. For more details regarding genes belonging to each cluster please refer to Fig. S3. Genes are not drawn to scale.

of the type $TTGATN_4ATCAA$ (named FNR binding box). Thus, the promoter regions of the four gene clusters were analyzed to identify the presence of these binding boxes, up to 200 bp upstream their ATG site. Homologous consensus sequences were identified based on its score and distance to the putative transcription site (Fig. 2). For most of the gene clusters, the putative binding regions are located around −10 to −35 bp, except for *dnr* (−88 bp). These distances are in agreement with the average location of these sites identified in the whole genome of *P. aeruginosa* (*Trunk et al., 2010*) and also with the more common spacing locations observed for FNR in *E. coli*, which are usually centered at approximately −41, −61, −71, −82 and −92 bp from the transcription initiation point (*Wing, Williams & Busby, 1995*). Although these binding sites still need to be experimentally confirmed, the involvement of MARHY0862 (FNR homologue) and MARHY3023 (DNR homologue) in the regulation of the denitrification pathway will be discussed.

The other regulatory system of denitrification is NarXL that responds to nitrate and is proposed to be involved in the regulation of the *nar* genes (*Schreiber et al., 2007*). NarL, the transcription regulator, recognizes specific heptameric motifs (TACYYMT) with a 7 − 2 − 7 arrangement located in the promoter regions of *nar* gene cluster (*Schreiber et*

*al., 2007*). The promoter region of *narK* and *narGHJV* was analyzed to identify the −10 and −35 boxes, which were only found upstream *narK*, which lead us to propose that *narKGHJV* is transcribed into a single polycistronic unit. Moreover, upstream *narK* a potential NarL binding site (around −21 bp, TACCCCA, with a score of 4.13 for NarL of *E. coli*) and a FNR/DNR binding site were identified but none on *narGHJV* (Fig. 2). Relative to *narXL*, no binding sites for FNR/DNR nor NarL were identified upstream the −10/−35 boxes, which can indicate that these genes are constitutive. Moreover, NarL is not involved in regulation of *dnr* as no consensus sequences were identified upstream of FNR box, as opposite to what was described for *P. stutzeri* (*Härtig et al., 1999*) (*dnrE*) and *P. aeruginosa* (*dnr*) (*Schreiber et al., 2007*) that carried the motifs and were shown to have a NarL-dependent transcription.

## Analysis of gene expression during *M. hydrocarbonoclasticus* 617 growth in the presence of nitrate - microoxic batch growth

The expression of denitrification genes of *M. hydrocarbonoclasticus* 617 grown in the presence of nitrate, in the 2 L-bioreactor, was analyzed by quantitative real-time PCR. The results obtained for both *norB* genes (*MARHY3054* and *MARHY3014*) indicates that *norB_MARHY3054* (*c-norB*) gene is the one transcribing an active nitric oxide reductase under the denitrifying conditions used here, as no significant expression levels are observed for the *norB_MARHY3014* (*q-norB*) gene in these conditions (Fig. 3). Thus, *q*-type NOR (encoded by *norB _MARHY3014*) does not contribute to the *M. hydrocarbonoclasticus* denitrification pathway, under these conditions, and its role in the physiology of this microorganism remains unknown.

Expression profiles show that the genes encoding the catalytic subunits of the denitrifying enzymes (*narG*, *nirS*, *norB_MARHY3054* and *nosZ*) are expressed simultaneously after 1 h, and the maximum level of transcription occurs at time-point 5 h (end of the first diauxic phase), for these four genes (Fig. 3).

At 5 h of growth, *narG* expression level is slightly lower than the other genes (Fig. 3), reflecting the existence of additional regulatory mechanisms in *nar* expression or a lower strength of the *fnr* promoter in the *nar* gene cluster.

## Transcriptional analysis of *nos* gene cluster

In *M. hydrocarbonoclasticus nos* cluster is composed of *nosRZDFYL* (see Fig. S2). The *nosR* and *nosZ* genes are separated by 75 bp, while 68 bp separate *nosZ* from *nosD*. The *nosD*, *nosF* and *nosY* genes overlap each other by 2 bp and finally, *nosY* and *nosL* are separated by 22 bp. This organization suggests that these genes might be transcribed into a single transcriptional unit. To investigate this hypothesis reverse transcription of total RNA, from a sample of the bioreactor, was performed and the intergenic regions of the corresponding cDNAs were amplified by PCR. Amplicons with the expected sizes were obtained from *nosR* and *nosZ*, *nosZ* and *nosD*, and *nosDFYL* intergenic regions (Fig. 4A and 4B), showing that *nosRZDFYL* is transcribed into a polycistronic unit, similarly to *P. aeruginosa nos* gene cluster (*Arai, 2003*).

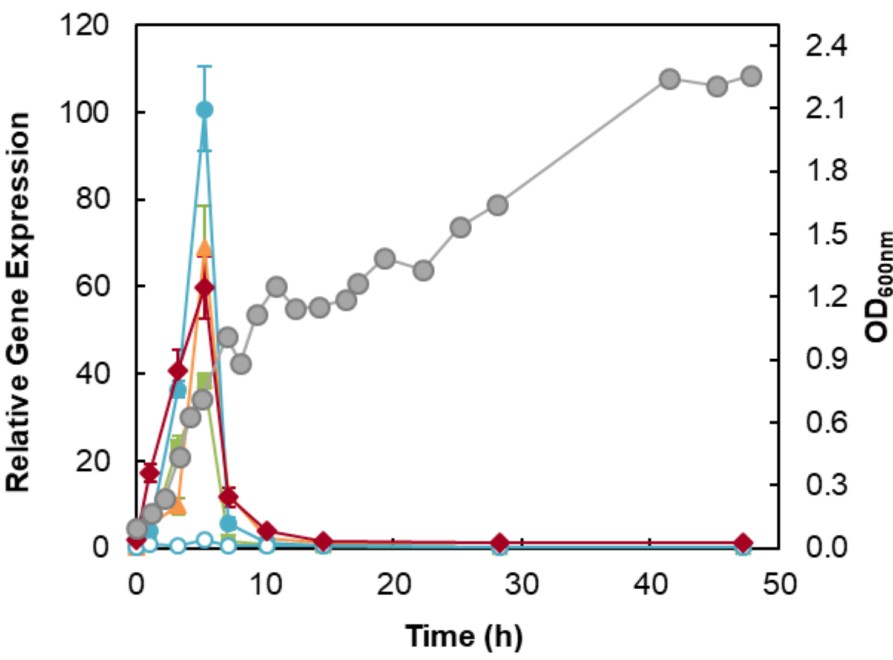

**Figure 3** **Profile of the expression levels of the genes encoding the catalytic domains of the denitrification enzymes.** Expression of *narG* (green squares), *nirS* (orange triangles), *c-norB* (blue filled circles), *q-norB* (blue open circles) and *nosZ* (red diamonds) encoding the catalytic domains of *M. hydrocarbonoclasticus* denitrification enzymes along the 50 h of growth under microaerobic conditions in the bioreactor. Relative expression values were obtained by normalizing expression of each target gene to the reference gene 16S rRNA, as described in Materials and Methods. Growth curve is represented as grey circles in the secondary axis.

### *Marinobacter hydrocarbonoclasticus* growth under denitrifying conditions

*M. hydrocarbonoclasticus* is a ubiquitous marine bacterium that populates the subtropical area and the Mediterranean Sea. In the present work, this bacterium was grown under two different conditions, microoxic and anoxic, as a model for the metabolism occurring at two different sea depths. The growth in the bioreactor was performed under microoxic conditions as a model for the layers close to the surface, while the growth in the sealed serum flasks are a model for deeper sea layers.

### Growth in a bioreactor under microoxic conditions

*M. hydrocarbonoclasticus* cultures were performed in a 2-L bioreactor under microoxic conditions (low aeration rate) and in the presence of nitrate (10 mM), to promote denitrification (*Moura et al., 2017*; *Eady, Antonyuk & Hasnain, 2016*; *Zumft, 1997*). The analysis of the growth curve shows the presence of a diauxic growth. The first growth phase occurs without any lag phase during the first 5 h with a growth rate of $0.173 \pm 0.008$ $h^{-1}$, slowing down (growth rate of $0.011 \pm 0.001$ $h^{-1}$) until approximately 35 h, when it reaches the stationary phase (Fig. 5A). The lag phase was not observed in any of the replicated assays, indicating
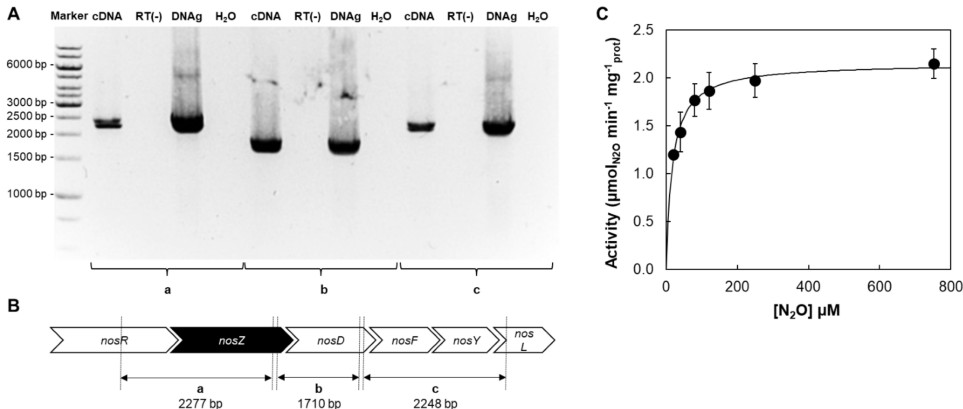

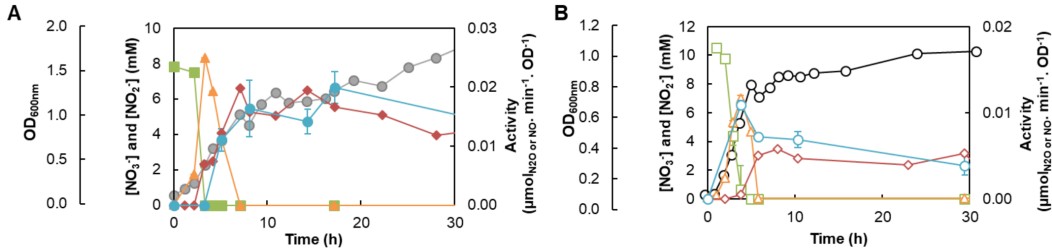

**Figure 4** *M. hydrocarbonoclasticus* **genome encodes a Clade I N$_2$OR.** (A) Analysis of the transcriptional organization of *M. hydrocarbonoclasticus* nitrous oxide reductase (*nos*) gene cluster. PCR products of intergenic regions between (a) *nosR-nosZ*, (b) *nosZ-nosD* and (c) *nosD-nosL* using cDNA (first lane), "RT-minus" control reaction (second lane), genomic DNA (third lane) and water (fourth lane). No amplifications were obtained in the "RT-minus" control reaction, indicating that the amplification from cDNA samples was not due to the presence of contaminating genomic DNA. (B) Genomic organization of the *nos* gene cluster in the genome of *M. hydrocarbonoclasticus*. The gene encoding for N$_2$OR is colored in black. (C) Kinetic activity for the reduction of N$_2$O by the whole-cells of *M. hydrocarbonoclasticus* grown under microaerobic conditions in the presence of nitrate at pH 7.5. Data were fitted to Michaelis-Menten equation, using a $K_m$ of $18 \pm 5\ \mu$M and $V_{max}$ of $2.2 \pm 0.1\ \mu$mol$_{N2O}$ min$^{-1}$ mg$^{-1}$ of total protein.

**Figure 5** *M. hydrocarbonoclasticus* **profile of denitrification pathway metabolites and enzyme activity.** Nitrate (green squares) and nitrite (orange triangles) concentrations (primary axis) and nitric oxide (blue circles) and nitrous oxide (red diamonds) reduction rates by the whole-cells determined at different time-points are shown in the secondary axis for the growth under microaerobic conditions in the bioreactor (A) or performed in sealed serum flasks (B). The activity is given as micromoles of NO or N$_2$O reduced per minute per optical density. The assays were performed as described in Materials and Methods, in triplicates. Three biological replicates were analyzed (for the data presented in this Figure) but not at the exact same time points, thus a representative experiment is shown (similar results were obtained for the other replicas). The growth curve of *M. hydrocarbonoclasticus* in the bioreactor (grey circles) and in the sealed serum flasks (black open circles) is included in the additional axis in A and B, respectively.

that adaptation-period was not required, or it was very small; and the death phase was not reached even after 48 h of growth, suggesting that the carbon source, lactate, had not been completely consumed.

The oxygen level was monitored during the growth, showing a rapid decrease in the first 2 h, and remaining negligible until the end of the growth (see Fig. S2). This apparent

absence of oxygen does not mean that there is no oxygen, but that the amount dissolved is being completely consumed: the low aeration rate combined with the low agitation at 50 rpm after 5 h of growth will lower the amount of dissolved oxygen in the growth media. This foresees that the microorganisms will grow in microenvironments experiencing anoxic or near anoxic conditions.

A diauxic growth was reported for *P. denitrificans* during anoxic growth in the presence of nitrate (60 mM) (*Hahnke et al., 2014*), with the first corresponding to the consumption of nitrate and the second with the start of nitrite consumption. However, in the present study the two exponential phases seem to correspond to the simultaneous use of oxygen and nitrate as terminal electron acceptors by *M. hydrocarbonoclasticus*, in the first phase, and in the sole use of the low amount of oxygen dissolved in the growth media in the second exponential phase, with the first having an expected energetic advantage.

### *Growth in sealed serum flasks*

*M. hydrocarbonoclasticus* was grown in sealed serum flasks in a medium with the same composition and under similar condition of agitation (in this case orbital shaking). Inspection of the growth curve indicates that the exponential phase starts one hour after inoculation and has a duration of 5 h, with a growth rate of $0.22 \pm 0.02$ $h^{-1}$ (Fig. 5B), as the first growth phase of the microoxic growth. The stationary phase occurs after this time-point and lasts until 50 h of growth, without observation of a death phase.

## Activity profile of denitrifying enzymes

Nitrate and nitrite were quantified during *M. hydrocarbonoclasticus* growth, to monitor the action of NaR and $cd_1$NiR, respectively, which catalyze the first and second steps of the denitrification pathway (Fig. 1). In addition, the reduction rate of nitric oxide and nitrous oxide by whole-cells was also determined at different time-points during growth, to identify when $c$-NOR and $N_2$OR were active in the cells (Fig. 5).

The data obtained indicate that nitrate was being consumed in the second hour of the growth and was completely consumed after 3 h (first mid-exponential phase of the diauxic growth and the time-point at which the oxygen level is negligible, see Fig. S1). This is an indication that nitrate transport to the cytoplasm and NaR were active before time-point 2 h in the bioreactor. Indeed, at time-point 1 h, *narG* gene expression was already occurring at low levels (Fig. 3), and thus it is likely that NaR was already active in the cells. Moreover, the decrease in nitrate concentration was concomitant with the formation of nitrite, which reaches a maximum after 3 h of growth with a value close to the initial nitrate concentration (8.5 mM), and four hours later (during the initial stages of the second diauxic growth phase) nitrite was completely consumed (Fig. 5A). This decrease in nitrite concentration is attributed to the activity of $cd_1$NiR. In fact, a correlation between *nirS* expression and nitrite consumption is observed (Fig. 3 and 5A). Moreover, similar nitrate and nitrite profiles were observed for the growth in the sealed serum flasks, indicating that both enzymes are also active under these conditions (Fig. 5B).

The last two steps of the denitrification pathway are catalyzed by $c$-NOR and $N_2$OR, and to determine if these enzymes are active in *M. hydrocarbonoclasticus* the reduction

rate of NO and $N_2O$ by the whole-cells was monitored (Fig. 5). The results obtained from the growth carried in the bioreactor indicate that both enzymes started to be active in the cells after 5 h of growth. Moreover, it was observed that these enzymes are active in the cells until the end of the growth (Fig. 5), even if NO and $N_2O$ are expected to have been completely consumed earlier on (*Bergaust et al., 2010*).

On the other hand, the profile of the reduction rate of NO and $N_2O$ by the whole-cells grown in the sealed serum flasks showed maximum NO reduction rate at 3 h, when high levels of nitrite are still present in the medium, while the $N_2O$ reduction was only observed after approximately 6 h of growth, indicating that both enzymes are active under these conditions at pH 7.5 (Fig. 5B). Moreover, contrary to what was observed for the cells cultured in the bioreactor under microoxic conditions, the ability to reduce these metabolites decreases towards the end of the growth.

### Kinetic parameters for $N_2O$ reduction by whole-cells

The $N_2O$ reduction by the whole-cells of *M. hydrocarbonoclasticus* as a function of substrate, $N_2O$, has a hyperbolic behavior (Fig. 4C). This data was fitted to the Michaelis–Menten equation to estimate the kinetic parameters, $V_{max}$ of $2.2 \pm 0.1\ \mu mol_{N2O}\ min^{-1}\ mg^{-1}$ and $K_m$ of $18 \pm 5\ \mu M$. This $K_m$ is of the same order of magnitude as the $K_m$ of Clade I *P. stutzeri* strain DCP-Os1 $N_2OR$ ($36 \pm 9\ \mu M$) determined by $N_2O$ consumption measurements, which are lower than those reported for whole-cells containing Clade II $N_2OR$ ($1.0 \pm 0.2\ \mu M$ and $1.3 \pm 0.4\ \mu M$, from *Dechloromonas denitrificans* strain ED-1 and *Anaeromyxobacter dehalogenans* strain 2CP-C, respectively) (*Yoon et al., 2016*).

## DISCUSSION

### Gene expression and regulation of denitrification in *M. hydrocarbonoclasticus*

In *M. hydrocarbonoclasticus* there is a 70 kb region that encodes all the genes involved in the denitrification pathway (see Fig. S3), with the promoter regions of these gene clusters presenting a FNR/DNR consensus sequences, while NarL binding sequences were only identified upstream the *nar* operon (Fig. 2).

The genome of this bacterium has genes that encode two different types of NORs, *q*-NOR and *c*-NOR, but under the denitrifying conditions used here only the later catalytic gene subunit is expressed at significant levels (Fig. 3), and in fact this was the enzyme isolated from *M. hydrocarbonoclasticus* membrane extracts (*Timoteo et al., 2011*).

In *M. hydrocarbonoclasticus*, the onset and maximum of *nosZ*, *nirS*, *narG* and *norB* transcription occurs simultaneously, which suggests that a common signal(s) triggers and regulates the expression of these genes. The triggering signal can be the oxygen, which levels are very low during the growth in the bioreactor (Fig. S1). This behavior is different from the one in *P. denitrificans*, in which it was observed that *nosZ* is expressed earlier than *nirS* and *norB* (*Bergaust et al., 2010*). Those authors interpreted this pattern of transcription by *nosZ* transcription being triggered by oxygen, while the onset of *nirS* and *norB* transcription was associated with the increase in nitric oxide concentration that occurs afterwards (*Bergaust et al., 2010*) (*vide infra*).

Relative to the regulation of *M. hydrocarbonoclasticus nar* genes, similarly to what was observed in *P. stutzeri* (*Härtig et al., 1999*) and *P. aeruginosa* (*Schreiber et al., 2007*), these might be under the regulation of *narXL*, a nitrate-responsive two-component system. This was supported by the identification of NarL DNA binding sites in the promoter region of *nar* gene cluster (Fig. 2), as mentioned.

The data obtained here does not univocally identified the signals involved in the regulation of *M. hydrocarbonoclasticus* denitrification genes. However, it is expected that the common effectors are oxygen and nitric oxide, as identified in other denitrifying bacteria (*Spiro, 2012*), through the action of FNR and DNR, respectively. In fact, given the pattern of transcription observed for the genes encoding the catalytic denitrifying subunit, it can be postulated that oxygen and nitric oxide are the main effectors in this organism. Upon decrease of oxygen tension in the cultures, FNR monomers with its [4Fe-4S] centers dimerize, becoming activate, and bind specific DNA regions, activating the transcription of several genes required for the anoxic metabolism, which include activation of *dnr* and *narGHJ* (*Crack et al., 2013*; *Trunk et al., 2010*). In turn, upon the initial activity of $cd_1$NiR, originated from the initial low levels of *nirS* transcription, NO concentration will increase by the reduction of nitrite, and DNR will activate *nir*, *nor* and *nos* genes (Fig. 6). On the other hand, *nar* genes would be regulated by oxygen and nitrate that activates their expression through *narXL*.

The existence of a cascade type regulation for these genes will prevent the accumulation of intermediate toxic products. Although, FNR and DNR recognize the same promoter regions (as mentioned before), it has been shown that DNR is the specific regulator of denitrification (*Ebert et al., 2017*; *Trunk et al., 2010*). The mechanism for the discrimination of these binding sites is still unknown, though nitrosylation of FNR could play a role, as upon nitrosylation (which might occur when nitric oxide concentration increases), FNR reduces its binding affinity to the specific DNA regions (*Crack et al., 2013*). This double regulatory mechanism might be important to avoid increasing release of toxic nitric oxide by NiR, together with down-regulation of *nir*, *nor* and *nos* gene clusters, as with a second regulator, as DNR, denitrification can proceed by activating those gene clusters.

## Metabolites and NO/$N_2O$ reduction rates by whole-cells

Nitrate and nitrite concentration during the growth of *M. hydrocarbonoclasticus* was monitored, being observed a rapid consumption of nitrate with concomitant formation of nitrite, which is completely consumed after 2 h, in both types of growth conditions tested. Considering that the decrease in nitrite concentration is due to $cd_1$NiR activity, as the denitrification pathway is active, the concomitant formation of nitric oxide is expected, followed by formation of $N_2O$. The activity of the two last enzymes of the denitrification pathway was proven by the observation of the NO and $N_2O$ reduction rate by the whole-cells, which occurs until the end of the growth (Fig. 5).

There was a delay between maximum gene expression (*nirS*, *norB* and *nosZ*) and enzymatic activity or complete nitrite consumption: (i) *nirS* maximum expression occurs at 5 h and complete consumption of nitrite occurs only after time-point 8 h; and (ii) *nosZ* and *norB* maximum expression occurs at 5 h, while ability of the cells to reduce NO and

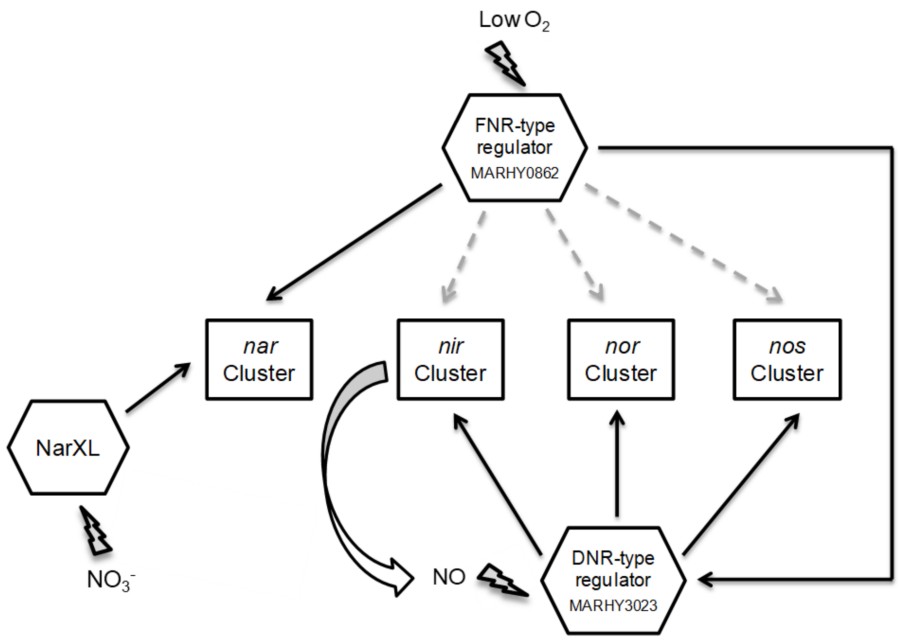

**Figure 6** **Putative regulatory network controlling denitrification gene expression in *M. hydrocarbono-clasticus*.** The main effector involved in regulation of nitrate reductase (*nar*), nitrite reductase (*nir*), nitric oxide reductase (*nor*) and nitrous oxide reductase (*nos*) gene clusters is the low oxygen tension through a FNR-type regulator. Secondary effectors are nitric oxide (NO) (first produced by the initial low levels of nitrite reductase) involved in expression regulation of *nir*, *nor* and *nos* genes clusters, through a DNR-type regulator, and nitrate ($NO_3^-$) involved in regulation of *nar* genes through the two-component system NarXL. Our own bioinformatic analyses suggest that MARHY0862 and MARHY3023 are, respectively, the FNR and DNR homologues involved in regulation of denitrification gene expression in *M. hydrocarbono-clasticus*. Dash lines indicate the putative but lower level of *nir*, *nor* and *nir* gene regulation by FNR.

$N_2O$ is maximum at 7 h. This displacement can be attributed to the maturation process of these three enzymes, as *c*-NOR is a membrane-bound enzyme and, $cd_1$NiR and $N_2$OR catalytic center assembly involves several accessory factors.

Comparison between the activity profiles obtained for the two types of growth showed that lower reduction rates, especial of $N_2O$, are obtained in the sealed serum flasks. Two major explanations can be hypothesized, either there is reduced transcription of *nosZ* genes under this condition or the amount of $N_2$OR produced is lower. In fact, the yield of $N_2$OR isolated from a batch bioreactor operated under microoxic conditions is $2.6 \pm 0.4$ mg/L medium, while from a similar growth under anoxic conditions is $0.5 \pm 0.3$ mg/L medium.

Nevertheless, since cells isolated from both growth conditions were able to reduce nitrous oxide, it is plausible to argue that *M. hydrocarbonoclasticus* can carry out the four steps of the denitrification pathway under these two growth conditions, and this organism has a denitrifying chain as the one schematically presented in Fig. 1.

## $N_2$OR Clade and nitrous oxide affinity

*M. hydrocarbonoclasticus* has an affinity for nitrous oxide consistent with the presence of a Clade I $N_2$OR, which also agrees with the arrangement and gene constitution of *nos* operon

in this organism, *nosRZDFYL* (Fig. 4B). Furthermore, the estimated $K_m$ for $N_2O$ reduction by the whole-cells ($18 \pm 5$ μM) (Fig. 4C) is similar to the value reported for the $N_2O$ reduction by the *in vitro* activated fully-reduced $N_2OR$ ($K_m$ of $14 \pm 3$ μM) (*Dell'acqua et al., 2008*).

Considering that 40 μL of growth under microoxic conditions has a reduction rate of $3.4 \times 10^{-2}$ $\mu mol_{N2O} min^{-1}$ (before being collected), and $1.49 \times 10^{-4}$ mg of $N_2OR$ (considering a 70% recovery for the isolation of $N_2OR$ from this growth), this gives an estimate of 228 $\mu mol_{N2O} min^{-1} mg_{N2OR}^{-1}$ for the cells *in vivo*. This value is in fact similar to the specific activity of $N_2OR$ in the activated fully reduced form (200 $\mu mol_{N2O} min^{-1} mg_{N2OR}^{-1}$) (*Johnston et al., 2014*). In the case of the anoxic growth, an activity of the same order of magnitude as that one was estimated at the end of the growth (340 $\mu mol_{N2O} min^{-1} mg_{N2OR}^{-1}$).

## CONCLUSIONS

*M. hydrocarbonoclasticus* presents in its genome all the genes encoding the catalytic subunits of enzymes responsible for each step of the denitrification pathway. These genes, as well as those encoding accessory factors, necessary for enzyme maturation, transcriptional regulators and substrate transporters, were found to be organized in a common *locus*. Genetic clustering of denitrification genes has been previously described for other bacteria, but clustering of genes encoding the four denitrifying reductases, as observed in *M. hydrocarbonoclasticus*, seems to be less frequent (*Demanèche et al., 2009*; *Philippot, 2002*). Such genomic arrangement of denitrification genes favors the hypothesis of the existence of "denitrification islands" that can be transferred horizontally across species (*Zumft, 1997*).

In the growth performed in the bioreactor under microoxic conditions in the presence of nitrate, the genes encoding the catalytic subunits of denitrifying enzymes have a maximum expression level at the end of the first diauxic phase (around 5 h, 2–3 h after oxygen levels reach its minimum value), which rapidly decreases to very low levels. Thus, we propose that the initial regulatory signal is oxygen, through a FNR-like transcriptional regulator (MARHY0862) that activates the DNR-type transcription regulator (MARHY3023). This transcription regulator responds to NO, a secondary signal, formed by NiR and that will regulate the expression of *nir*, *nor* and *nos* clusters. The other secondary regulatory effector is nitrate, involved in the regulation of *nar* genes expression, through the two-component system NarXL (Fig. 6).

In both type of growths, nitrate consumption occurs in the early hours of the growth, while nitrite consumption, and the ability of the whole-cells to reduce nitric and nitrous oxide occurs later (at 5 h of growth). This delay can be attributed to the requirement in accessory factors for the maturation of these enzymes to produce an active enzyme.

Moreover, our data indicate that *c*-NOR and $N_2OR$ are active until the end of the growth (48-50 h) in both type of growths. Thus, a functional denitrification pathway in *M. hydrocarbonoclasticus* is observed from nitrate to dinitrogen gas under these growth conditions.

*M. hydrocarbonoclasticus* has a Clade I $N_2OR$, as the whole-cells have $18 \pm 5$ μM affinity for $N_2O$, corroborating the gene composition and organization of *nos* operon, which is

transcribed as polycistronic unit, and it was estimated that the whole-cells present an activity per amount of $N_2OR$ that is consistent with the specific activity of the isolated enzyme upon activation.

*M. hydrocarbonoclasticus* is one of the major species that populates the subtropical oceans and the Mediterranean Sea. The metabolism is modulated by the availability of oxygen in these facultative anaerobes, with the oxygen tension decreasing with the sea depth. Therefore, to mimic two natural growth conditions (lower sea depth and surface sea layers), this organism was grown under microoxic (bioreactor) and anoxic (sealed serum flasks) conditions. Our data shows that under both conditions the enzymes that catalyze the four steps of denitrification are active. These data and methodology will be used in the future to monitor other effects on the denitrification pathway of this organism, such as temperature and pH.

### Funding

This work was financially supported by Fundação para a Ciência e Tecnologia through the projects PTDC/BIA-PRO/098882/2008 (Sofia R. Pauleta) and PTDC/BIA-PRO/109796/2009 (Sofia R. Pauleta), and the scholarship SFRH/BD/87898/2012 (Cíntia Carreira). This work was also supported by the Unidade de Ciências Biomoleculares Aplicadas-UCIBIO, which is financed by national funds from FCT/MEC (UID/Multi/04378/2013) and co-financed by the ERDF under the PT2020 Partnership Agreement (POCI-01-0145-FEDER-007728). The funders had no role in the study design, data collection and analysis, decision to publish, or preparation of the manuscript.

### Grant Disclosures

The following grant information was disclosed by the authors:
Fundação para a Ciência e Tecnologia: PTDC/BIA-PRO/098882/2008, PTDC/BIA-PRO/109796/2009, SFRH/BD/87898/2012.
FCT/MEC: UID/Multi/04378/2013.
ERDF: POCI-01-0145-FEDER-007728.

### Competing Interests

The authors declare there are no competing interests.

### Author Contributions

- Cíntia Carreira performed the experiments, analyzed the data, prepared figures and/or tables, authored or reviewed drafts of the paper, approved the final draft.
- Olga Mestre performed the experiments, analyzed the data, authored or reviewed drafts of the paper, approved the final draft.
- Rute F. Nunes performed the experiments, authored or reviewed drafts of the paper, approved the final draft.
- Isabel Moura authored or reviewed drafts of the paper, approved the final draft.

# PeerJ

- Sofia R. Pauleta conceived and designed the experiments, analyzed the data, contributed reagents/materials/analysis tools, prepared figures and/or tables, authored or reviewed drafts of the paper, approved the final draft.

## Data Availability

The raw data are provided in the Supplementary Files.

## Supplemental Information

Supplemental information for this article can be found online at http://dx.doi.org/10.7717/peerj.5603#supplemental-information.

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
