# Peer review of "Genomic organization, gene expression and activity profile of Marinobacter hydrocarbonoclasticus denitrification enzymes"

_PeerJ, doi:10.7717/peerj.5603_

## Round 0.1 · original submission · Major Revisions

Both reports we received advise publication subject to suitable changes. I find most requests to be extremely reasonable and probably relatively easy to address. Contrary to the first comment of reviewer #2, I do not think the manuscript suffers from excessive length.

·

Basic reporting

The text is in general well written, although it may be improved, as some parts of the text is hard to understand.

In Introduction, there are some parts that can be deleted and others that can be expanded. Some references can be added or be upto date.

Raw data have to include all data from replicates, not only one.

The authors did not respect the format for the Abstract as defined in the Author guides, as well as the interlined spaces (should have been 2 line space).

Some figures can be deleted or moved in the Supplemental data

See my comments in the Report document

Experimental design

Most of comments related to this criterion are included in my Report document. In general, this criterion is partially covered.

Validity of the findings

Most of comments related to this criterion are included in my Report document. In general, this criterion is partially covered

Additional comments

My general and specific comments are provided in an attached document

Reviewer 2 ·

Basic reporting

The paper is too long and could be significantly shortened.

Experimental design

No comment.

Validity of the findings

No comment.

Additional comments

In this paper, the authors interrogate the genome of Marinobacter hydrocarbonoclasticus to draw conclusions about the biochemical apparatus for denitrification, and the regulation of the corresponding genes. Perhaps most interesting is the coding potential for a Q-type NO reductase, though the authors' data suggest that only the C-type enzyme is expressed and active under the growth conditions that they use.

The organism grows rapidly by denitrification and qRT-PCR data show essentially simultaneous expression of the denitrification genes during the rapid phase of growth. Bioreactor cultures show a second phase of slower growth, it is less clear what is going on physiologically during this second phase.

From the presence of regulatory genes and putative cis-acting regulatory sequences the authors infer a likely regulatory network governing the expression of the denitrification genes. There are no major surprises here and regulation seems to follow the same general patterns found in other organisms (that is, with major roles for FNR- and NarXL-type regulators).

Specific comments

1. Lines 366-368. Here the authors are comparing distance from the start codon to distance from the transcription start site, which does not really make sense. Distance from the start codon is not a very useful parameter.

2. Lines 377 and 379. Here distances from the transcription start site are given, but I think the authors mean distances from the start codon (according to the Figure).

3. Lines 401-403. This may just reflect differences in absolute promoter strength rather than regulatory mechanisms.

4. Lines 425 and 436. I would avoid mentions of death phase since viable counts have not been measured. Dead cells contribute to optical density.

5. Lines 514-521. Some literature citations are needed in this passage.

6. There is scope to shorten the paper, for example by avoiding repetition. Some observations are repeated in the results, discussion and conclusions.

---

## Round 0.2 · Minor Revisions

Please address the few remaining issues (especially the transcription levels time-course)

·

Basic reporting

Clear and unambiguous, professional English used throughout.
OK

Literature references, sufficient field background/context provided.
OK

Professional article structure, figs, tables. Raw data shared.
OK

Self-contained with relevant results to hypotheses.
OK

Experimental design

Original primary research within Aims and Scope of the journal.
OK

Research question well defined, relevant & meaningful. It is stated how research fills an identified knowledge gap.
OK

Rigorous investigation performed to a high technical & ethical standard.
OK

Methods described with sufficient detail & information to replicate.
See the attachment

Validity of the findings

Data is robust, statistically sound, & controlled.
Not perfect, but satisfactory


Conclusion are well stated, linked to original research question & limited to supporting results.
OK

Speculation is welcome, but should be identified as such.
OK

Additional comments

See the attached doument

---

## Round 0.3 · accepted · Accept

Thank you for addressing the remaining issues.

#